# Longitudinal analysis of the rectal microbiome in dogs with diabetes mellitus after initiation of insulin therapy

**Nicole L. Laia**[1¤], **Patrick C. Barko**[1], **Drew R. Sullivan**[1], **Maureen A. McMichael**[2], **David A. Williams**[1], **Jennifer M. Reinhart**[1]*

**1** Department of Veterinary Clinical Medicine, College of Veterinary Medicine, University of Illinois at Urbana-Champaign, Urbana, Illinois, United States of America, **2** Department of Clinical Sciences, College of Veterinary Medicine, Auburn University, Auburn, Alabama, United States of America

¤ Current address: Virginia Veterinary Centers, Richmond, Virginia, United States of America
* jreinha2@illinois.edu

**Data Availability Statement:** All sequence files are available from the NCBI BioProject database (accession number PRJNA723003).

## Abstract

There have been numerous studies in humans and rodents substantiating the role of the gastrointestinal microbiome in the pathogenesis and progression of both type 1 and type 2 diabetes mellitus. Diabetes mellitus is a common endocrinopathy in dogs; however, little is known about the composition of the gut microbiome during the development and treatment of diabetes in this species. The objective of this pilot study was to characterize the gastrointestinal microbiome of dogs with diabetes mellitus at the time of diagnosis and over the first 12 weeks of insulin therapy and identify associations with glycemic control. Rectal swabs and serum for fructosamine measurement were collected from 6 newly diagnosed diabetic dogs at 2-week intervals for 12 weeks. Rectal samples were sequenced using 16S, ITS, and archaeal primers. Measures of alpha and beta diversity were assessed for changes over time; associations between absolute sequence variant (ASV) relative abundances and time and fructosamine concentration were identified using a microbiome-specific, multivariate linear effects model. No statistically significant changes over time were noted in alpha diversity and samples significantly grouped by dog rather than by time in the beta diversity analysis. However, multiple ASVs were negatively (*Clostridium sensu stricto 1*, *Romboutsia*, *Collinsella*) and positively (*Streptococcus*, *Bacteroides*, *Ruminococcus gauveauii*, *Peptoclostridium*) associated with time and two ASVs were positively associated with fructosamine (*Enterococcus*, *Escherichia-Shigella*). These changes in gastrointestinal microbial composition warrant further investigation of how they may relate to diabetes mellitus progression or control in dogs.

## Introduction

In health, the intestinal microbiome plays a key role in a diverse array of physiologic processes including nutrient absorption, energy metabolism, immune regulation, and maintenance of the gastrointestinal barrier [1–4]. Conversely, alterations in the microbiome, termed dysbiosis,

 

**Funding:** This study was funded by the Illinois Canine and Feline Clinical Research Grant Program (JR, DW, MM, PB, DS), administered by the University of Illinois, College of Veterinary Medicine (vetmed.illinois.edu). The funders had no role in study design, data collection and analysis, decision to publish, or preparation of the manuscript.

**Competing interests:** The authors have declared that no competing interests exist.

are associated with a variety of diseases including diabetes mellitus (DM) [5, 6]. Current evidence suggests that dysbiosis both contributes to the development of DM and interferes with its successful management through a variety of mechanisms.

Inflammation is proposed as a pivotal link between the gut microbiome and DM. In these patients, dysbiosis is generally characterized by an increase in opportunistic pathogens which generate low-level inflammation and stimulate production of inflammatory mediators. At the same time, there is a decrease in bacterial populations that produce short-chain fatty acids (SCFAs), such as butyrate, which are essential to gut epithelial health and immunotolerance to normal gut flora. Therefore, decreased butyrate production leads to impaired mucosal barrier function and further perpetuates the systemic effects of gut inflammation [6]. The gut microbiome has additional effects on host glycemic control. SCFAs, indoles, and other mediators stimulate release of incretin hormones, which are a major determinant of post-prandial insulin release. Modulation of the bile acid pool by gut microbes also influences incretin secretion and alters hepatic glucose and glycogen handling. Additionally, the enteric microbiota influences the function of adipose tissue, which plays an important role in metabolic regulation. Thus, there appears to be a complex relationship between the gut microbiome and glycemic status [7].

In type 1 DM, dysbiosis and gut inflammation are suspected to stimulate the immune system, which may contribute to or trigger autoimmune destruction of the endocrine pancreas [5]. This theory is supported by progressive changes in markers of dysbiosis in children with type 1 DM over time from before seroconversion to the development of auto-antibodies to overt hyperglycemia [5]. Furthermore, early manipulation of the intestinal microbiome has been shown to delay onset of type 1 DM in mouse models [8]. In type 2 DM, inflammatory mediators from low-level gut inflammation are thought to induce insulin resistance contributing to both the development and perpetuation of hyperglycemia [6]. Several studies have documented the presence of dysbiosis in type 2 diabetics including associations between relative species abundances and markers of diabetic regulation such as fasting blood glucose and hemoglobin a1c [9–12].

Despite the growing body of evidence in human medicine, the role of the microbiome in canine DM is relatively unknown. DM is a common endocrinopathy in dogs with reported prevalence of approximately 0.34–1.2% [13]. Although not typically classified as type 1 or type 2, dogs have characteristics of both with an absolute insulin deficiency due to pancreatic islet cell destruction and concurrent factors promoting insulin resistance and complicating regulation [14]. Given the similarities between human and canine DM, it is possible that canine DM is also associated with dysbiosis. To date only one study has evaluated this hypothesis. This cross-sectional study found that diabetic dogs did not have a statistical difference in alpha diversity in comparison to a control group of dogs, although they did have an increased relative abundance of *Enterobacteriaceae* compared with healthy controls [15]. However, this study looked only at a single time point and included dogs at various stages of treatment and diabetic control; therefore, no inferences can be made about alterations in the microbiome at time of diagnosis or transitions during treatment. Thus, the impact of DM treatment on the gastrointestinal microbiome in canine diabetics remains unknown.

The study described herein was a longitudinal, prospective, pilot study evaluating the rectal microbiome of newly diagnosed diabetic dogs throughout the first 12 weeks of treatment. The primary objective of this study was to characterize the gastrointestinal microbiome of dogs with DM at the time of diagnosis and during treatment. We hypothesized that initially there would be changes in diversity and composition of the rectal microbiome of diabetic dogs from the time of diagnosis through the first 12 weeks of treatment. Specifically, we hypothesized that alpha diversity would increase with treatment time and that relative abundances of

opportunistic pathogens, such as Enterobacteriaceae, would decrease, while butyrate-producing organisms, such as Firmicutes, would increase as treatment proceeded. A secondary objective was to evaluate abundances of microbial taxa relative to patient fructosamine, as a quantitative marker of diabetic control. We hypothesized that there would be significant correlations between fructosamine and relative abundance of one or more taxa in this study.

## Materials and methods

### Animals

Newly diagnosed diabetic dogs were recruited through partnering primary care veterinary clinics. The dogs were presented for clinical signs related to their disease. At the time of diagnosis, all dogs had a complete blood count, serum biochemistry profile, urinalysis, urine culture, serum total thyroxine, fructosamine, cobalamin, folate concentrations, trypsin-like immunoreactivity (TLI), and pancreatic lipase immunoreactivity (PLI) performed to rule out concurrent disease. Dogs were excluded if diagnostics supported concurrent disease, such as urinary tract infection, exocrine pancreatic insufficiency, or neoplasia. Dogs with increased TLI, PLI, or liver enzymes were not excluded. Additionally, dogs that received antibiotics within 4 weeks prior to enrollment or during the study period and dogs that had a diet change in the 4 weeks prior to enrollment or during the study period were excluded. All dogs were confirmed to be eating a nutritionally balanced, commercial dog food and that they were eating that same diet for at least a month prior to study start. Dogs were maintained on their preinvestigation diet throughout the study. If admitted dogs did require antibiotics, developed illness, or had a change in diet, they were excluded from further participation, but samples collected prior to the protocol deviation were included in the analysis. The diabetic management, including insulin choice and dosage, were at the discretion of the primary care veterinarian.

Dogs were evaluated every 2 weeks for the first 12 weeks of insulin therapy at which time weight was recorded, serum was collected for fructosamine measurement, rectal swabs were obtained, and questionnaires by the owner and primary care veterinarian were completed (S1 File). The questionnaires included information on breed, weight, body and muscle condition scores, sex, age, diet, medication history, appetite, water intake and urination, activity level, insulin dose and type, and subjective evaluation of control. The protocol was approved by the University of Illinois Institutional Animal Care and Use Committee (Protocol #18043).

### Rectal swab sample collection and handling

Rectal swabs, serum, and urine were collected by the primary care veterinarians. Sterile swabs (Copan FLOQSwab; Copan Inc., Murrieta, CA) were inserted into the rectum and swept in a circular motion. The swabs were placed into a dry container. Samples were either shipped the same day or stored overnight at 4˚C. Samples were shipped overnight on ice. Once received serum and urine were processed same day and the rectal swabs were stored at -80˚C until DNA extraction was performed.

### DNA extraction

DNA was extracted from rectal swabs using QIAamp PowerFecal Pro Kit DNA (QIAGEN, Hilden, Germany) following the manufacturer's protocol with the exception that rectal swabs were used in the initial bead beating step rather than feces. To assess the extent of microbial contamination in swabs and reagents, we also processed a sterile (unused) swab and 200 μL of nucleotide-free, sterile water in place of rectal swabs in the DNA extraction protocol. DNA

concentration in the extracts was determined using a NanoDrop 1000 spectrophotometer (ThermoFisher Scientific, Waltham, MA) with quality check by agarose gel electrophoresis.

## Library preparation and sequencing

Rectal swab samples were submitted for relative quantification of eubacteria, archaea, and fungi at the University of Illinois Biotechnology Center. In addition to the rectal swab samples, we submitted the two negative controls from the DNA extraction and an additional negative control consisting of sterile, nucleotide-free water. First, the sequencing library was prepared using the Fluidigm Access Array system (Fluidigm Corporation, South San Francisco, CA) using the following primers: V3_F357_N (5-CCTACGGGNGGCWGCAG-3) and V4_R805 (5-GACTACHVGGGTATCTAATCC-3) for the V3-V4 region of eubacterial 16S rRNA; Arch349F (5-GYGCASCAGKCGMGAAW-3) and Arch806R (5-GGACTACVSGGGTATC TAAT-3) for archaeal 16S rRNA; and ITS3 (5-GCATCGAATGAAGAACGCAGC-3) and ITS4 (5-TCCTCCGCTTATTGATATGC-3) for the internal transcribed spacer 2 region of fungal species. Sequencing was performed on the Illumina MiSeq platform (Illumina, Inc., San Diego, CA) using 250 bp paired end reads.

## Bioinformatic processing of sequences

Processing of raw sequences was performed using the Quantitative Insights into Microbial Ecology software (version 2) [16]. Sequences were first demultiplexed to remove sample-specific index sequences. The Dada2 plugin was used to generate an absolute sequence variant (ASV) feature table [17]. Taxonomy of ASVs were assigned using the SILVA reference database (release 138.1) [18]. The demultiplexed and aligned sequences were further processed using the phyloseq R package [19]. ASVs with an unassigned phylum were removed and the remaining ASVs were filtered to exclude low-abundance taxa using a 50% prevalence threshold. Finally, the filtered count matrix was agglomerated on the taxonomic level of genus.

## Statistical analysis

Demographic and clinical data are presented descriptively. All amplicon sequencing data was analyzed in the R language for statistical programming (version 4.0.3; Bunny-Wunnies Freak Out) [20]. The Shannon diversity index (SDI), an estimate of alpha diversity, was calculated for each sample using the raw ASV count matrix [21]. Mixed-effects generalized linear models were used to assess the effect on SDI of sample collection time after insulin administration.

Unsupervised methods were used to detect changes in microbiota community structure associated with different time points following the initiation of insulin therapy. Beta diversity was estimated using the Bray-Curtis dissimilarity matrix generated using the filtered, agglomerated ASV count matrix and visualized with non-metric multidimensional scaling (NMDS). We used permutational multivariate analysis of variance (PERMANOVA) to detect statistical differences in microbial community structure among the individual dogs and sample collection time points. PERMANOVA was implemented on the Bray Curtis dissimilarity matrix with 1,000 permutations using the adonis function in the vegan package [22].

Microbiome Multivariable Association with Linear Models (MaAsLin2) was used to detect differentially abundant ASVs among sample collection time points and associations among ASVs and serum fructosamine concentrations [23]. The ASV count matrix was normalized by the total sum scaling method (TSS) and log transformed prior to analysis. To understand the impact of multiple testing, an estimate of the false discovery rate (FDR) was calculated using the Benjamini-Hochburg method [23, 24]. We reported results as statistically significant when the raw p-value was $< 0.05$, regardless of the FDR.

## Results

### Population

Six dogs were included in the study with one dog lost to follow up after week 6. Four were males and 2 females, all neutered. Mean age was 7.3 years (range 7.7–11) and mean weight was 16 kg (range, 4.2–42.6). The dog breeds represented were one each Chihuahua, Australian Cattle dog, Labrador retriever, Brussels Griffon, Yorkshire terrier mix, and Pomeranian (Table 1). All dogs were presented to their primary care veterinarian with the chief complaint of polyuria and polydipsia and one dog each additionally was noted to have lost weight and developed blindness. As per primary care records, all dogs had lost weight since examination prior to diagnosis of diabetes mellitus, but five of six dogs were overweight. Five out of six dogs were treated with NPH insulin and one dog was treated with porcine insulin zinc suspension (Vetsulin, Merck Animal Health, De Soto, KS).

### Clinical parameters of diabetes mellitus control

Diabetic control was assessed objectively by monitoring serum fructosamine and weight and subjectively by assessment of water intake, urinary output, and appetite. There was an overall downward trend in fructosamine over time and weight appeared stable throughout treatment (Fig 1). Most dogs also had stable to normalized water intake and urination as assessed by owners and all had improved subjective control, as assessed by the primary veterinarian, from time of diagnosis at week 0 to study completion at week 12 (Fig 2).

### 16S-rRNA sequencing statistics

A total of 3,581,921 sequences were generated from 2036 unique ASVs. Excluding negative controls, the median sequencing depth was 96,462 reads per sample (9,192–146,597). To understand the extent of environmental and reagent contamination, we sequenced three negative control samples. We generated 9,340 total sequences from 7 unique ASVs in the three negative control samples, nearly all of which were from the phylum Proteobacteria including three from the genus *Escherichia-Shigella* and one from the genus *Azomonas*. Sequences from *Escherichia-Shigella* represented >89% of ASVs in all three negative control samples. After removing the negative control samples, filtering out low-abundance sequences, and agglomerating the ASV counts on genus 2,649,900 sequences from 29 unique ASVs remained with a median depth of 73,224 (6962–110,625) sequences per sample.

### Alpha and beta diversity

The Shannon diversity index (SDI) was calculated for each dog at each time point and compared across time and between dogs. There were no significant differences in SDI over time

**Table 1. Patient characteristics at inclusion.**

| Patient ID | Breed | Age (years) | Sex | Weight (kg) | BCS (1–9) | Insulin type |
|---|---|---|---|---|---|---|
| 5 | Chihuahua | 8 | FS | 4.8 | 6.5 | NPH |
| 8 | Australian cattle dog | 8 | FS | 24.1 | 6 | porcine zinc suspension |
| 9 | Labrador Retriever | 11 | MN | 42.6 | 6 | NPH |
| 11 | Brussels Griffon | 11 | MN | 9.8 | 5 | NPH |
| 15 | Yorkshire terrier mix | 9 | MN | 10.5 | 6 | NPH |
| 16 | Pomeranian | 7 | MN | 4.2 | 4 | NPH |

BCS = body condition score; NPH = neutral protamine Hagedorn; FS = female spayed; MN = male neutered.

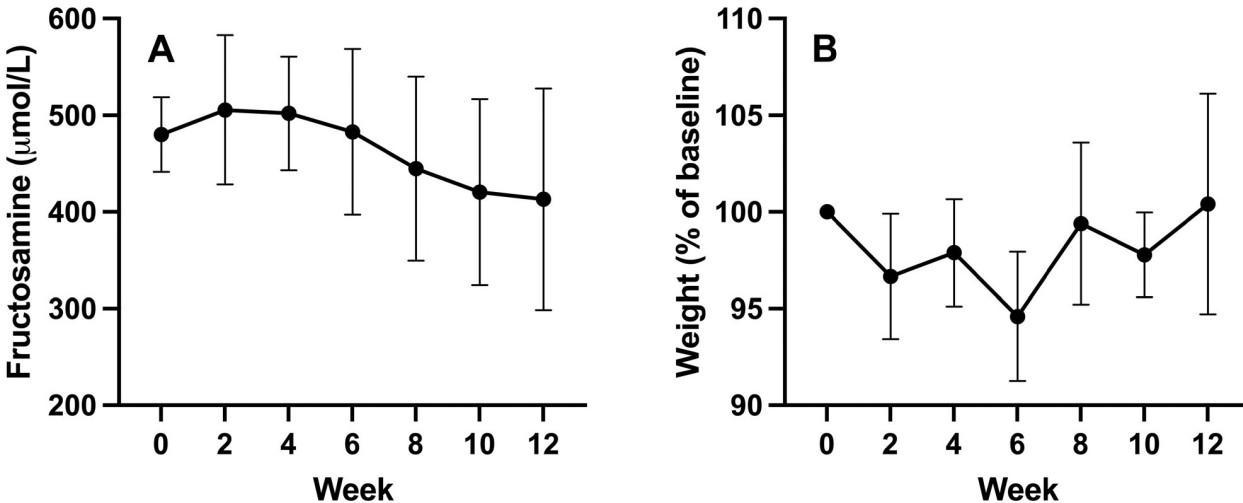

**Fig 1. Quantitative clinical measures of diabetic control.** Changes in (A) serum fructosamine concentrations and (B) % difference in weight from baseline (week 0) over time.

(p = 0.32; Fig 3). Non-metric multi-dimensional scaling (NMDS) of the Bray Curtis dissimilarity matrix (Fig 4) and PERMANOVA revealed that the origin of the sample with respect to an individual dog was a significant source of variation in microbial communities ($R^2 = 0.46$;

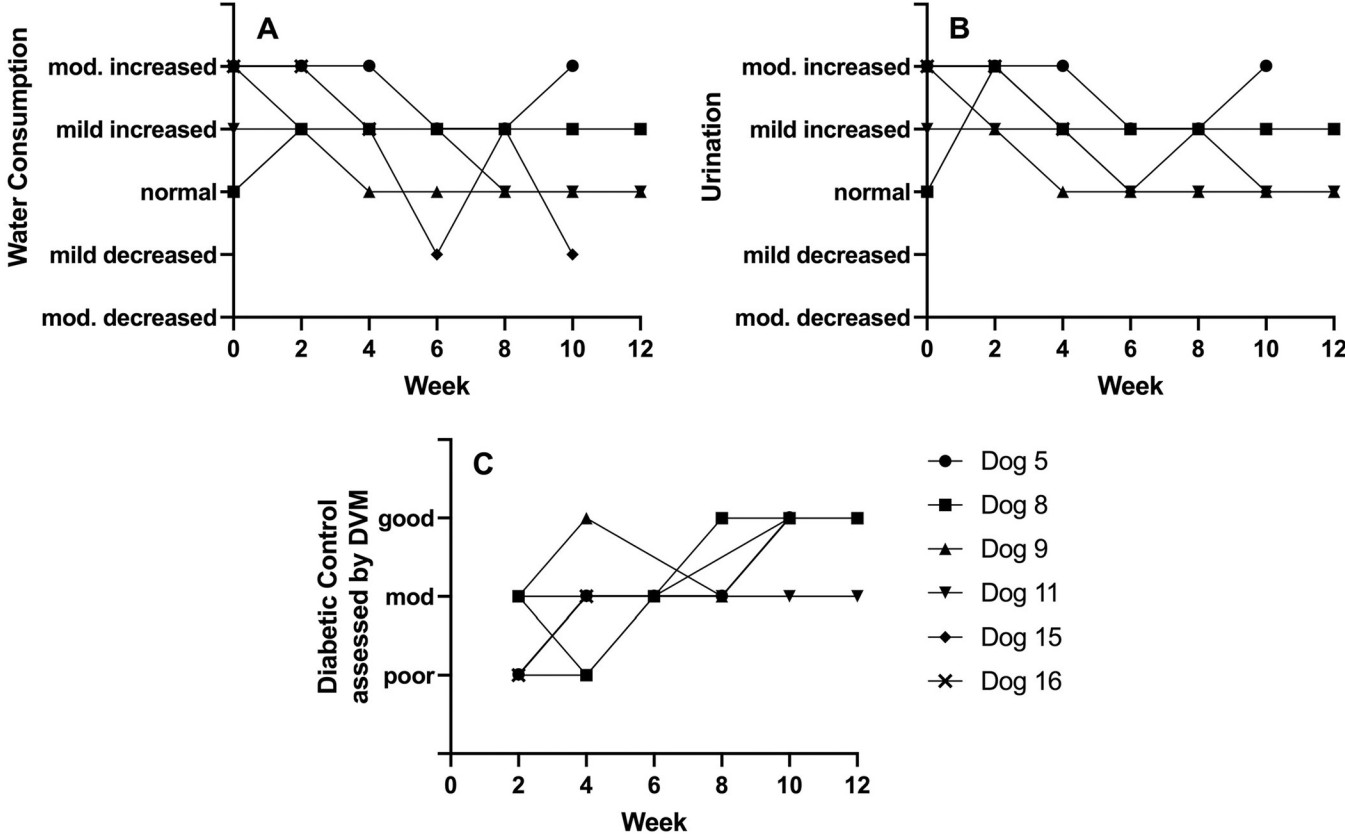

**Fig 2. Subjective clinical measures of diabetic control.** Subjective assessment of diabetic control over time including (A) water consumption assessed by owner, (B) urinary output assessed by owner, and (C) overall diabetic control assessed by primary veterinarian. mod = moderate.

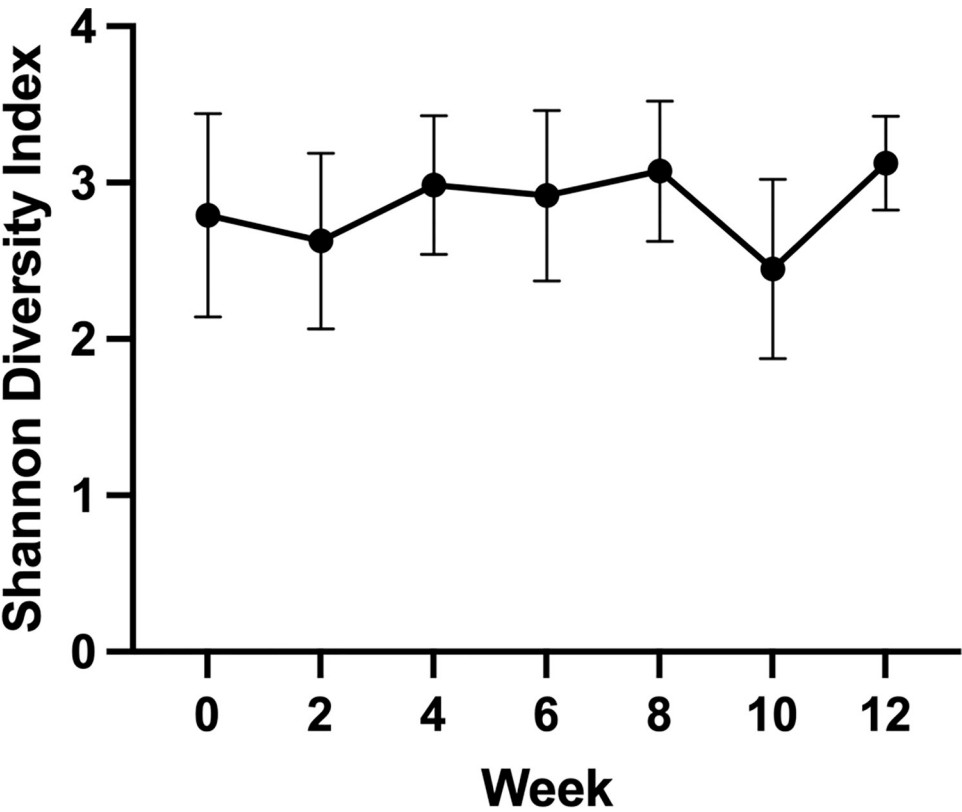

**Fig 3. Shannon diversity index.** Mean +/- standard deviation Shannon diversity index. There was no significant difference over time (p = 0.32).

P = 0.01). There was separation of samples from two dogs (patient ID 8 and 11) collected prior to insulin administration (week 0) collected after initiation of insulin therapy, but collection time point was not a statistically significant source of variation in community structure based on PERMANOVA ($R^2$ = 0.02; P = 0.39).

### Multivariate association of differential abundance, time, and fructosamine

The relative abundances of each ASV were assessed for changes over time and in association with fructosamine using MaAsLin2. Several ASVs did change in relative abundance over time (Table 2, Fig 5) and in relation fructosamine (Table 3, Fig 6).

### Archaea and fungi

Most amplicons from archaea primers mapped eubacterial species and only 3 true archaea ASVs were detected, all of which were isolated from a single dog. Similarly, the ITS primers detected only 3 fungal taxa, all belonging to the *Malassezia* genus. The ITS primers also yielded an amplicon from a nematode, *Passalarus ambiguus*.

### Discussion

In human medicine, there is a growing body of evidence suggesting that the gut microbiome contributes to the development and perpetuation of DM. Dysbiosis is consistently recognized across multiple studies of human diabetics and is associated with both islet cell autoimmunity

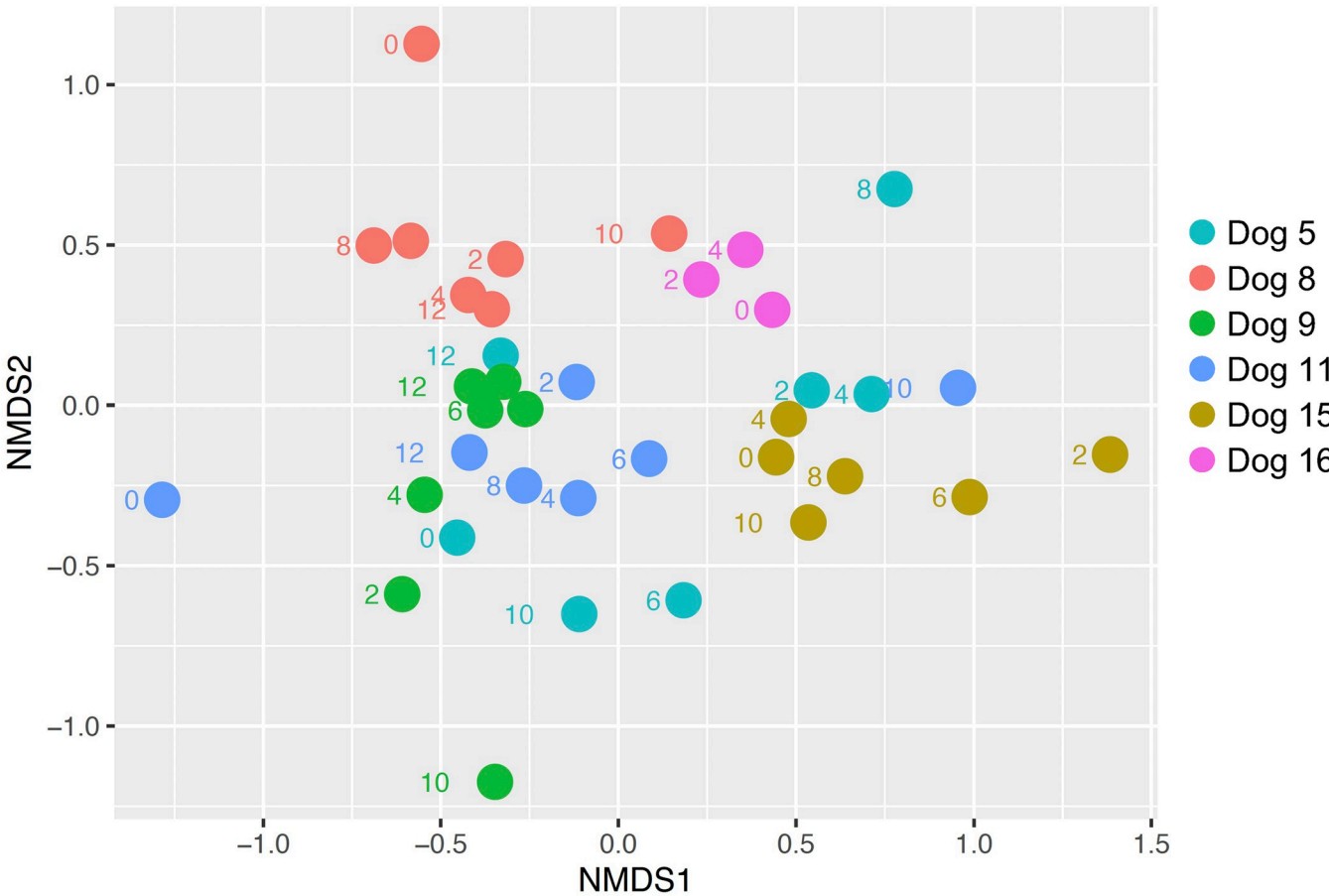

**Fig 4. Beta diversity analysis.** Non-metric multi-dimensional scaling of the Bray-Curtis dissimilarity index. Numbers next to each circle represent the week of sampling. The major source of variation among samples was the individual dogs ($R^2 = 0.46$; $P = 0.01$).

and insulin resistance [5, 6]. Additionally, therapeutic alterations to the microbiome using probiotics, fecal microbiota transplantation, and dietary modifications have been shown to enhance glucose metabolism [1]. Thus, understanding and manipulating the microbiome in DM has the possibility to improve clinical outcomes for diabetic patients. In contrast, little is known about the state of the gut microbiome in canine DM. Similar microbial-targeting therapies might also improve glycemic control in dogs, but a better characterization of DM-associated dysbiosis is first required in these patients. In this pilot study, we identified several ASVs in the gut microbiome of new diagnosed diabetic dogs that changed in relative abundance over time during the first 12 weeks of insulin therapy. Additionally, we identified two ASVs whose relative abundances positively correlated with serum fructosamine concentration, a commonly used marker of glycemic control in veterinary medicine.

In the present study, several bacterial taxa increased in relative abundance over the first 12 weeks of insulin therapy. Of these, *Bacteroides* was of particular interest. *Bacteroides spp*. have been previously reported to have positive effects on glucose tolerance and insulin sensitivity [6]. This finding is consistent across studies in human diabetics. In case-control studies, *Bacteroides spp*. relative abundance is generally decreased in diabetics relative to non-diabetic patients, and, in interventional studies, relative abundance increases with diabetic therapies [9, 25–27]. *Bacteroides spp*. are some of the better described butyrate-producing bacteria and promote colonocyte health and tight junction integrity. Thus, improved insulin sensitivity is

**Table 2. ASVs with significant relative abundance associations with various time points.**

| ASV | Week | Model Coefficient | p-value | FDR |
|---|---|---|---|---|
| *Clostridium sensu stricto 1* | 2 | -1.653 | 0.008 | 0.255 |
| | 4 | -1.506 | 0.014 | 0.389 |
| | 6 | -2.121 | 0.002 | 0.087 |
| | 8 | -2.213 | 0.001 | 0.087 |
| | 10 | -2.251 | 0.001 | 0.087 |
| | 12 | -2.085 | 0.003 | 0.133 |
| *Romboutsia* | 2 | -1.416 | 0.022 | 0.413 |
| | 4 | -1.246 | 0.042 | 0.451 |
| | 6 | -1.509 | 0.021 | 0.413 |
| | 12 | -1.655 | 0.019 | 0.413 |
| *Collinsella* | 4 | 1.151 | 0.039 | 0.451 |
| | 8 | 1.198 | 0.043 | 0.451 |
| *Streptococcus* | 2 | 0.920 | 0.047 | 0.466 |
| | 4 | 0.960 | 0.039 | 0.451 |
| *Bacteroides* | 12 | 1.497 | 0.038 | 0.451 |
| *Ruminococcus gauveauii* | 8 | 1.150 | 0.029 | 0.451 |
| *Peptoclostridium* | 8 | 0.880 | 0.032 | 0.451 |

ASV = absolute sequence variant; FDR = false discovery rate.

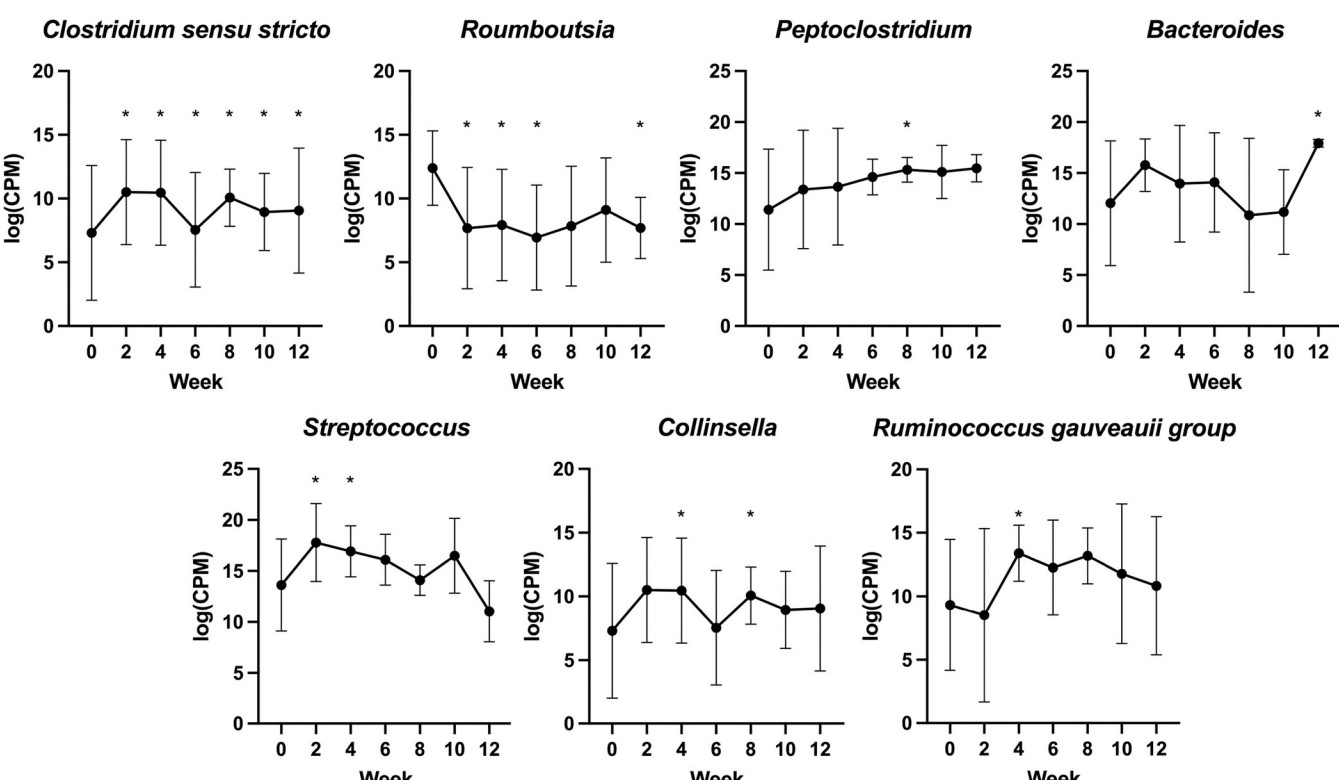

**Fig 5. Changes in relative abundances of ASVs with significant associations with time.** * indicates relative abundance significantly different from week 0.

**Table 3. ASVs with relative abundance correlated with fructosamine.**

| ASV | Correlation coefficient | P-Value | FDR |
|---|---|---|---|
| *Enterococcus* | 0.757 | 0.001 | 0.018 |
| *Escherichia-Shigella* | 0.575 | 0.007 | 0.099 |

ASV = absolute sequence variant; FDR = false discovery rate.

thought to occur via reduced gut permeability, LPS production, and systemic inflammation. It is possible that a similar mechanism is present in canine diabetics, which might have contributed to the overall improved clinical status of the animals in the present study.

We also identified two ASVs that decreased in relative abundance over the study period, *Roumboutsia* and *Clostridium sensu strictu 1*; these were decreased from baseline at several timepoints. In humans with type 2 DM, increased *Roumboutsia* relative abundance has been associated with poor glucose tolerance and reduce fasting insulin levels [28]. Thus, a decrease in relative abundance over the first 12 weeks of DM treatment may represent improved glycemic control in our study dogs. In contrast, *Clostridium sensu strictu 1* relative abundance is generally decreased in human type 2 and gestational DM patients [11, 29, 30]. Given this finding, relative abundance might be expected to increase with better diabetic regulation, rather than decrease, as was observed in the present study. However, the association between *Clostridium sensu strictu 1* and metabolic status has primarily been documented in cross-sectional studies comparing diabetics to healthy controls, rather than longitudinal studies, so direct comparison between these and the present study may not be applicable. Alternatively, this disparity could represent species-related differences in the gut microbiome and/or differences in DM pathogenesis between humans and dogs.

Our analysis also evaluated for correlations between ASV relative abundance and serum fructosamine concentrations. Fructosamine is an indirect measurement of serum protein glycosylation, which increases during chronic hyperglycemia [13]. It is commonly used in combination with clinical signs as a measure of diabetic control in canine patients. Two bacterial taxa, *Enterococcus* and *Escherichia-Shigella* were positively associated with serum fructosamine

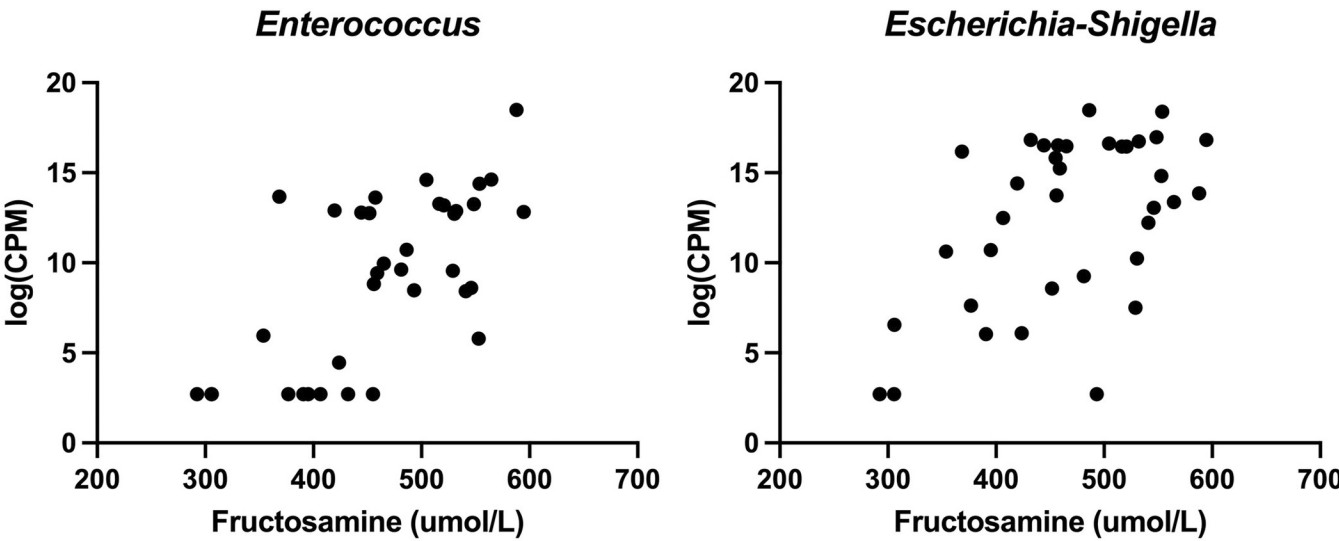

**Fig 6. Fructosamine vs. relative abundance of ASVs associated with fructosamine concentration.**

in this study. The latter taxa, *Escherichia-Shigella*, is particularly interesting as the relative abundance of the family *Enterobacteriaceae*, of which *Escherichia-Shigella* are members, has been documented to be increased in diabetic dogs compared to healthy controls [15]. It is important to note that *Escherichia-Shigella* is a common contaminant in lab reagents and sequences represented >89% of ASVs in all 3 negative controls in our study. Therefore, the abundance of *Escherichia-Shigella* in the subject samples might have been falsely elevated [31, 32]. Despite this, we expect contamination from lab reagents to be consistent throughout all samples, so positive association in with serum fructosamine concentration likely represents true biologic processes rather than contamination.

Although we identified changes in relative abundances of several ASVs, we did not identify any significant changes in alpha or beta diversity. Measures of alpha diversity (e.g., Shannon diversity index) and beta diversity (e.g., Bray Curtis dissimilarity index) detect large shifts in diversity within and between microbial populations, respectively, but are insensitive to changes in individual bacterial taxa. Thus, it is possible to have significant and important shifts in key bacterial relative abundances without affecting measures of diversity. In fact, this phenomenon occurs commonly in studies investigating the gut microbiome of human diabetic patients [6]. Lack of changes in measures of diversity in our study could be due to several different factors. Firstly, our study was designed as a pilot study and therefore had limited enrollment. It is possible that more dramatic changes in diversity would have been noted with more dogs or with a longer time course. Also, it is possible that some dogs' DM may not have been adequately controlled by week 12 of treatment, which could have made changes in diversity associated with glycemic control difficult to detect. Insulin type also varied between dogs, which could have obscured changes in diversity over time. It is interesting to note that only one dog was treated with porcine insulin zinc suspension, while the others received NPH. This dog had a more precipitous decline in fructosamine and subjectively the veterinarian assessed the dog to be well controlled. Finally, it is possible alterations in the gut microbiome during initial diabetic therapy in dogs are driven more by changes in relative abundance of a few key taxa rather than overarching measures of diversity.

Most microbiome research is focused on the bacterial composition, but fungal and archaeal organisms are often not explored even though they are estimated to comprise 1% and 0.01–0.3% of the microbial cells in fecal samples, respectively [33]. Given that all fungal ASVs generated in this study were from genus *Malassezia*, we suspect that the organisms identified originated from the skin rather than the rectum due to the collection method. There is not a universal collection method that provides a representative sample of the entire gastrointestinal tract, but most researchers use fecal samples as they are easily and non-invasively collected. Our results could have been different had voided fecal samples been used but we selected rectal swabs to maintain consistency between multiple centers and ensure samples were collected from every visit. The finding of *P. ambiguus* is unexpected as this is a species-specific, rabbit pinworm that is not known to infect dogs and we speculate that it represents ingestion of pinworm DNA that was passing through the dog's GI tract.

An important limitation of the differential abundance analysis in this study is the high FDR among significant results. Many similar studies use the FDR as a threshold for statistical significance, accepting results with FDR-adjusted p-values ranging from <0.05 to <0.25. The authors acknowledge that this would be a more robust approach to assigning statistical significance, compared with the unadjusted (raw) p-values. As this was a pilot study, we sought to strike a balance between avoiding false discoveries and missing the discovery of actual differences that could inform future targeted investigations. Thus, we have provided the estimates of the FDR for all significant results so the reader can understand the likelihood that a given

result is a false discovery. We emphasize that our results should be confirmed in future studies using larger samples sizes and targeted approaches to quantify microbial communities.

This pilot study was designed to characterize changes in the gastrointestinal microbiome in dogs with DM during the first 12 weeks of treatment, not to assign causality between the intestinal microbial composition and host health. However, these results do demonstrate shifts within the intestinal microbiome of dogs with diabetes during initial insulin therapy. These trends provide justification for future research in this area including prospective studies into long-term changes within the microbiome and how modifying the microbiome with therapies such as probiotics or fecal microbial transplantation may affect diabetic management in dogs. Additionally, investigation of the interaction between insulin and dietary therapy on the microbiome of diabetic dogs is essential. In the present study, we maintained dogs on their pre-diagnosis diet to better isolate the effect of insulin therapy. However, many veterinarians do recommend a diet change as part of diabetic management. The impact of combination diet change and insulin therapy on the gastrointestinal microbiome could have significant implications for future therapies of difficult to regulate diabetic dogs.

## Conclusions

This study evaluated gut microbiome alterations in newly diagnosed diabetic dogs with the goal of identifying changes correlated with time and fructosamine. While statistically significant changes in diversity were not observed, several bacterial species did have shifts in relative abundances associated with time and serum fructosamine concentrations. The changes identified here warrant further investigation to improve our understanding of the role of the gastrointestinal microbiome in canine diabetes mellitus, which could lead to future targeted treatment strategies.

## Supporting information

**S1 File. Owner and veterinarian questionnaires.** Separate questionnaires were developed for the first (time 0) and subsequent (time 2–12) visits to assess owner and veterinarian perceptions of control of clinical signs of diabetes mellitus.
(DOCX)

## Acknowledgments

The authors thank Drs. Jason Verbeck, Auldon Francis, and Patty McElroy for assistance with patient recruitment and sample collection. The authors also thank Dr. Christopher Fields for assistance with data analysis.

## Author Contributions

**Conceptualization:** Patrick C. Barko, Maureen A. McMichael, David A. Williams, Jennifer M. Reinhart.

**Data curation:** Nicole L. Laia, Patrick C. Barko, Jennifer M. Reinhart.

**Formal analysis:** Nicole L. Laia, Patrick C. Barko, Jennifer M. Reinhart.

**Funding acquisition:** Patrick C. Barko, Drew R. Sullivan, Maureen A. McMichael, David A. Williams, Jennifer M. Reinhart.

**Investigation:** Nicole L. Laia, Patrick C. Barko, Drew R. Sullivan, Jennifer M. Reinhart.

**Methodology:** Nicole L. Laia, Patrick C. Barko, Maureen A. McMichael, David A. Williams, Jennifer M. Reinhart.

**Project administration:** Nicole L. Laia, Patrick C. Barko, Drew R. Sullivan, Jennifer M. Reinhart.

**Resources:** Nicole L. Laia, Patrick C. Barko, Jennifer M. Reinhart.

**Software:** Patrick C. Barko.

**Supervision:** Patrick C. Barko, Drew R. Sullivan, David A. Williams, Jennifer M. Reinhart.

**Writing – original draft:** Nicole L. Laia, Patrick C. Barko, Jennifer M. Reinhart.

**Writing – review & editing:** Nicole L. Laia, Patrick C. Barko, Drew R. Sullivan, Maureen A. McMichael, David A. Williams, Jennifer M. Reinhart.

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
