## [Decision Letter · Decision Letter 0]

12 Jul 2022

PONE-D-22-09246Longitudinal analysis of the rectal microbiome in dogs with diabetes mellitus after initiation of insulin therapyPLOS ONE

Dear Dr. Reinhart,

Thank you for submitting your manuscript to PLOS ONE. After careful consideration, we feel that it has merit but does not fully meet PLOS ONE’s publication criteria as it currently stands. Therefore, we invite you to submit a revised version of the manuscript that addresses the points raised during the review process.

We look forward to receiving your revised manuscript.

Kind regards,

Jasbir Singh Bedi

Academic Editor

PLOS ONE

Journal Requirements:

Reviewers' comments:

Reviewer's Responses to Questions

**Comments to the Author**

1. Is the manuscript technically sound, and do the data support the conclusions?

Reviewer #1: Yes

2. Has the statistical analysis been performed appropriately and rigorously? 

Reviewer #1: Yes

3. Have the authors made all data underlying the findings in their manuscript fully available?

Reviewer #1: Yes

4. Is the manuscript presented in an intelligible fashion and written in standard English?

Reviewer #1: Yes

5. Review Comments to the Author

Reviewer #1: Line 91 Enterobacteriaceae should be italicized

Line 216 As should be added before per

Line 264 “was a large and significant” should be reframed

Line 267 “from” to be removed

Line 282 Please add full stop

Line 320 Instead of Diabetic it should be Insulin

Line 353 Check spelling of Enterobacteriaceae

Line 359 Please see “in”

Line 381 Replace Methods with method

Line 398 Studies.

There are few questions:

1. The authors said

“The objective of this pilot study was to characterize the gastrointestinal microbiome of dogs with diabetes mellitus at the time of diagnosis and over the first 12 weeks of insulin therapy and identify associations with glycemic control”

However they are also suggesting that

“This pilot study was not designed to make inferences about cause and effect between the intestinal microbial composition and host health; however, it does demonstrate shifts within the intestinal microbiome of dogs with diabetes during initial insulin therapy”.

So, if, the objective was not to make any inference of cause and effect between the intestinal microbial composition and host health then what is the purpose of this study?

2. In this study statistically significant changes in microbial diversity were not observed thus it means that the microbiome did not change significantly before and after the initiation of the insulin therapy. Does it mean that there is no significant role of microbial diversity in DM in dogs.

6. PLOS authors have the option to publish the peer review history of their article (what does this mean?). If published, this will include your full peer review and any attached files.

Reviewer #1: No

---

## [Author Response · Author response to Decision Letter 0]

18 Jul 2022

Many thanks to the reviewer for their comments and questions. We have done our best to address all concerns, both in the manuscript and below.

Line 91 Enterobacteriaceae should be italicized – corrected

Line 216 As should be added before per – added

Line 264 “was a large and significant” should be reframed – “large and” removed

Line 267 “from” to be removed – removed

Line 282 Please add full stop – period removed

Line 320 Instead of Diabetic it should be Insulin – corrected

Line 353 Check spelling of Enterobacteriaceae – corrected and italicized

Line 359 Please see “in” – I’m sorry I don’t understand this correction

Line 381 Replace Methods with method – corrected

Line 398 Studies. – I don’t understand this correction either

Thank you for the above corrections. There were a couple (lines 359 and 398) that I didn’t understand. For some reason, the line numbers provided don’t match up with my version of the proof. If you could provide further context and more specific recommendations, I would be happy to address them.

There are few questions:

1. The authors said

“The objective of this pilot study was to characterize the gastrointestinal microbiome of dogs with diabetes mellitus at the time of diagnosis and over the first 12 weeks of insulin therapy and identify associations with glycemic control”

However they are also suggesting that

“This pilot study was not designed to make inferences about cause and effect between the intestinal microbial composition and host health; however, it does demonstrate shifts within the intestinal microbiome of dogs with diabetes during initial insulin therapy”.

So, if, the objective was not to make any inference of cause and effect between the intestinal microbial composition and host health then what is the purpose of this study?

Thank you, this is a point that deserves clarification. In our study, we characterized the GI microbial changes in our population during initial treatment for DM. However, because our study design was single-arm longitudinal (it would be unethical to include an untreated control group), we cannot say definitively that the changes we observed were due to treatment. However, the changes in relative abundance that we found are consistent with what would be expected as glycemic control improves, based on studies in people and mice. Therefore, some relationship between microbial composition and diabetic status seems likely but cannot be definitively stated based on our results and warrants future study. I have clarified this point in the discussion (lines 414-417).

2. In this study statistically significant changes in microbial diversity were not observed thus it means that the microbiome did not change significantly before and after the initiation of the insulin therapy. Does it mean that there is no significant role of microbial diversity in DM in dogs.

Another point that deserves clarification, thank you. It is an incorrect, albeit common, assumption that if there are no changes in measures of microbial diversity, then there are no changes in microbial composition. Measures of alpha and beta diversity broadly assess intra- and inter-individual diversity, respectively. They are an attempt to identify and explain large shifts in microbial populations due to investigated factors. However, measures of diversity are insensitive to changes in individual bacterial taxa, which often have important effects on the host. This is why we also assessed the relative abundance of bacterial taxa (analogous to ASVs) and found several genera/species that demonstrated significant changes over time and correlations with fructosamine (see Tables 2 and 3). So, in short, lack of significant changes in measures of diversity does not mean there are no changes within the microbiome. I have added language to the discussion to clarify this point for readers (lines 365-375).

---

## [Editor Report · Decision Letter 1]

16 Aug 2022

Longitudinal analysis of the rectal microbiome in dogs with diabetes mellitus after initiation of insulin therapy

PONE-D-22-09246R1

Dear Dr. Jennifer

We’re pleased to inform you that your manuscript has been judged scientifically suitable for publication and will be formally accepted for publication once it meets all outstanding technical requirements.

Kind regards,

Jasbir Singh Bedi

Academic Editor

PLOS ONE
---

## [Editor Report · Acceptance letter]

26 Aug 2022

PONE-D-22-09246R1 

Longitudinal analysis of the rectal microbiome in dogs with diabetes mellitus after initiation of insulin therapy 

Dear Dr. Reinhart:

I'm pleased to inform you that your manuscript has been deemed suitable for publication in PLOS ONE. Congratulations! Your manuscript is now with our production department. 

Kind regards, 

on behalf of

Dr. Jasbir Singh Bedi 

Academic Editor

PLOS ONE